# The Effects of Dietary Protein Level on the Growth Performance, Body Composition, Intestinal Digestion and Microbiota of *Litopenaeus vannamei* Fed *Chlorella sorokiniana* as the Main Protein Source

**DOI:** 10.3390/ani13182881

**Published:** 2023-09-10

**Authors:** Hang Yuan, Wanlin Song, Jianqiang Tan, Yudong Zheng, Hongming Wang, Lili Shi, Shuang Zhang

**Affiliations:** 1College of Fisheries, Guangdong Ocean University, Zhanjiang 524088, China; yh17817895172@163.com (H.Y.); swling0313@163.com (W.S.); 17620111806@163.com (J.T.); yd_zheng96@126.com (Y.Z.); wanghongming97@163.com (H.W.); 2Key Laboratory of Aquatic, Livestock and Poultry Feed Science and Technology in South China, Ministry of Agriculture, Zhanjiang 524088, China; 3Aquatic Animals Precision Nutrition and High Efficiency Feed Engineering Research Center of Guangdong Province, Zhanjiang 524088, China; 4Guangdong Provincial Key Laboratory of Aquatic Animal Disease Control and Healthy Culture, Zhanjiang 524088, China

**Keywords:** *Litopenaeus vannamei*, *Chlorella sorokiniana*, growth performance, body composition, intestinal health

## Abstract

**Simple Summary:**

The shortage of fishmeal is currently a pressing issue for the aquaculture industry, as it increases the cost of aquatic product production and contributes to overfishing in the oceans. To solve this problem, *Chlorella sorokiniana* has started to be used in recent years as a major protein source in aquatic animal diets. However, differences in dietary protein sources will result in different optimal protein requirements. Accordingly, this study determined the growth performance, body composition, intestinal digestion and microbiota of *Litopenaeus vannamei* fed with five isolipid diets using *C. sorokiniana* as the main protein source but in different protein levels. The results showed that a dietary protein level of 40.81% resulted in the best weight gain rate when *C. sorokiniana* was used as the main protein source. Additionally, the optimal nutritional composition, digestibility and intestinal microbiota stability of *L. vannamei* occurred at a 40% dietary protein level. Higher dietary protein levels increased the colonisation of beneficial bacteria and inhibited the colonisation of pathogenic bacteria. These data inform the scientific application of *C. sorokiniana* in *L. vannamei* diets and promote the sustainable development of *L. vannamei* farming.

**Abstract:**

This study investigated the effect of dietary protein levels on *Litopenaeus vannamei*. Five isolipid diets with protein levels of 32%, 36%, 40%, 44% and 48% were prepared using *C. sorokiniana* as the main protein source. *L. vannamei* (initial body weight 0.83 ± 0.02 g) were fed these five diets for 8 weeks and referred to as the CHL32, CHL36, CHL40, CHL44 and CHL48 groups, respectively. When the feeding trial was finished, the growth performance, body composition, intestinal digestion and microbiota of *L. vannamei* were studied. The results showed that the maximum weight gain rate (WGR) of *L. vannamei* was in the CHL40 group while the lowest feed conversion ratio (FCR) was in the CHL48 group. According to the regression analysis using WGR as the evaluation index, the best growth performance of *L. vannamei* was obtained when the dietary protein level was 40.81%. The crude protein content of whole shrimp showed an increasing and then decreasing trend with increasing dietary protein levels. Furthermore, the *L. vannamei* muscle amino acid composition was relatively stable and, to some extent, independent of dietary protein levels. Trypsin, lipase and amylase (AMS) activity increased and then decreased with increasing dietary protein levels and, significantly, peaked in the CHL44 group. Analysis of the alpha diversity of the intestinal microbiota showed that the Chao1 index peaked in the CHL40 group and was significantly lower in the CHL48 group. Additionally, the relative abundance of pathogenic bacteria decreased significantly while the relative abundance of beneficial bacteria increased significantly in the intestine of *L. vannamei* as the dietary protein levels increased. The functional prediction of the intestinal microbiota revealed that dietary protein levels may influence the growth of *L. vannamei* by regulating various metabolic activities, and the highest WGR in the CHL40 group may have been related to the significant enrichment of nicotinate and nicotinamide metabolism and biotin metabolism functions. In summary, the optimal protein requirement for *L. vannamei* was around 40% when *C. sorokiniana* was used as the primary protein source. Too high or too low dietary protein levels could adversely affect shrimp body composition, intestinal digestion and microbiota.

## 1. Introduction

After decades of rapid growth, aquaculture is now the fastest-growing animal food production sector in the world. In 2022, global aquaculture production (87.5 million tons) surpassed marine capture (78.8 million tons) and became the main source of aquatic animals for human consumption [1]. Aquaculture provides a large source of high-quality protein for human beings and is also an important supplement for world food security, having made significant contributions to poverty eradication and increasing farmers’ income [2,3,4]. *Litopenaeus vannamei* is a representative species in aquaculture. It has a rapid growth rate, disease resistance, high muscle yield, delicious taste and rich nutrition, giving it a high economic value [5,6,7]. Compared to livestock animals, dietary protein plays a more important role in safeguarding health and promoting the growth of aquatic animals due to their low ability to utilise sugars [8]. Moreover, aquatic animals usually consume proteins to obtain essential and non-essential amino acids. As well as forming the basis for substances such as muscles and enzymes, proteins also provide energy through the tricarboxylic acid cycle, and thus, proteins are essential to the growth of aquatic animals [9,10]. Protein is generally considered to be irreplaceable, and the most important and expensive component of dietary nutrients for aquatic animals [11]. 

Protein is important for aquatic animals; so, are higher protein diets always better? No. Excess protein in the diet increases the burden of nitrogen excretion for aquatic animals, which pollutes farmed water bodies, limits the growth of farmed animals and greatly increases the cost of aquaculture [12]. Therefore, determining an appropriate level of dietary protein is important, both biologically and economically. However, due to the diversity of proteins in terms of amino acid composition, molecular weight and three-dimensional structure, proteins of different origins differ in their physicochemical properties. Because of this, the impact of changes in protein levels varied when different sources of protein were used [13]. Fishmeal (FM), a traditional protein source, has always been used as the main protein source in aquatic feed because of its high protein content, rich content of essential amino acids and vitamins, good palatability, low anti-nutritional factors, and ease of digestion and utilisation by aquatic animals [14]. Generally, the average amount of FM added to aquatic feed is higher than 30% and accounts for 40–60% of the cost of farming [15,16]. In recent years, the global production of FM has been declining year by year because of environmental pollution, overfishing and the El Niño phenomenon, with the latter exacerbating the imbalance between the supply and demand of FM, leading to a further sharp rise in FM prices and ultimately, limiting the rapid and sustainable development of the aquaculture industry [17,18,19]. Therefore, it is important to find and develop suitable new protein sources to replace FM in aquatic feed and to determine the optimal protein addition level of these new sources. This will have both economic and ecological significance. Specifically, on the economic side, higher economic returns can be obtained with lower feed costs, promoting the sustainable development of the aquaculture industry. On the ecological side, it would protect the marine environment and avoid damage to the marine ecosystem caused by overfishing for FM production [16,20]. 

At present, FM substitution in aquafeeds is mostly by plant protein sources—such as soybean meal (SBM) —and animal protein sources—such as meat and bone meal (MBM) and poultry by-product meal (PBM) [21,22]. However, plant proteins are very limited, as FM alternatives because of their lack of many essential amino acids, the presence of a large number of anti-nutritional factors and their low digestibility [23,24]. Animal proteins are a more suitable choice because they have a more complete amino acid composition, no anti-nutritional factors and better palatability. However, the quality of animal protein depends to a large extent on the source of raw materials, freshness and processing conditions, which are not well guaranteed due to the low prices [25]. Microalgae are a promising alternative protein source for future aquafeeds. Microalgae have a good amino acid structure, high protein and lipid content, and are rich in antioxidants and anti-immune stress compounds. More importantly, microalgae alternatives to FM are highly sustainable because microalgae are inexpensive and eco-friendly, simple to cultivate and reproduce quickly [26,27]. *Chlorella* was one of the first microalgae to be commercialised because of its high protein and lipid content, extremely rapid growth rate and its many trace elements, minerals and *Chlorella* growth factors (CGF) [28]. *C. sorokiniana* is one of the most used *Chlorella* strains in large-scale industrial production today [29]. Several studies have shown that *C. sorokiniana* strains are highly adaptable and can be cultured in a variety of aqueous environments such as freshwater, brackish and saline water using photoautotrophic, heterotrophic and mixed nutrient methods [30,31,32,33]. Additionally, several more in-depth studies have shown that a variety of microalgae, such as *C. sorokiniana*, have excellent nutrient uptake capacity and can effectively absorb nutrients from a variety of ‘wastewater streams’ including eutrophic water bodies, food industry effluent, agricultural wastewater streams, industrial wastewater streams and municipal wastewater streams [34,35,36]. Furthermore, *C. sorokiniana* is highly resistant to disease and stress and can absorb nutrients and efficiently produce valuable bioproducts in a wide range of wastewater streams that contain pathogens, heavy metals and various organic pollutants. All these properties imply that the development of microalgae such as *C. sorokiniana* may be able to simultaneously fulfil a variety of ecological and economic needs such as wastewater treatment, the synthesis of valuable metabolites and the accumulation of algal biomass as a raw material for animal diets [37,38,39]. However, due to the immaturity of current production technology and the lack of a complete industrial system, the production cost of *C. sorokiniana* as a protein source for aquatic animal diets is still high. As its production process gradually matures, it is likely that the great ecological and economic potential of *C. sorokiniana* will be gradually realised in the foreseeable future [34]. The replacement of FM by *Chlorella* protein has been studied in zebrafish [28], *Carassius auratus* [40], *Macrobrachium rosenbergii* [41], *Clarias gariepinus* [42] and *Micropterus salmoides* [43]. *Chlorella* protein has been shown to promote growth and reproduction, increase immune and digestive enzyme activity, and improve fatty acid composition in cultured animals after substitution. Currently, the use of *Chlorella* as a dietary component has also been applied to *L. vannamei*. A study by Pakravan et al. found that the addition of 97.2 g *Chlorella* per kg of diet significantly increased the growth performance, body fatty acid content and digestive enzyme activity of *L. vannamei* [44]. However, there are no relevant studies on the effects of changes in protein levels on *L. vannamei* when *Chlorella* is the main dietary protein source.

Accordingly, the present study explored the effects of dietary protein levels on the growth performance, body composition, intestinal digestion and microbiota of *L. vannamei* using *C. sorokiniana* powder (CHL) as the main protein source. The results provide a theoretical reference for the application of *C. sorokiniana* as a new protein source in aquatic feeds and to promote the sustainable farming of *L. vannamei*.

## 2. Materials and Method

### 2.1. Diet Preparation

Five isolipidic diets with protein levels of 32, 36, 40, 44 and 48% were configured using CHL as the main protein source (Table 1). CHL was obtained from the Institute of Hydrobiology, Chinese Academy of Sciences. Specifically, the CHL strains were first inoculated into modified Endo medium (30 g·L^−1^ glucose, 3 g·L^−1^ KNO_3_, 1.2 g·L^−1^ KH_2_PO_4_, 1.2 g·L^−1^ MgSO_4_•7H_2_O, 0.2 g·L^−1^ trisodium citrate, 0.016 g·L^−1^ FeSO_4_•7H_2_O, 2.1 mg·L^−1^ EDTA-Na_2_, 0.03 g·L^−1^ CaCl_2_•2H_2_O, 2.86 mg·L^−1^ H_3_BO_3_, 0.222 mg·L^−1^ ZnSO_4_•7H_2_O, 1.81 mg·L^−1^ MnCl_2_•4H_2_O, 0.021 mg·L^−1^ Na_2_MoO_4_, 0.07 mg·L^−1^ CuSO_4_•2H_2_O) and the initial pH of the modified Endo medium was adjusted to 6.0 with a 3 M NaOH solution. The mixture was incubated on an orbital shaker for 5–6 days at 30 °C and 180 rpm. The cultured parental algal strains were then inoculated into a 7.5 L fermenter (BIOFLO & CELLIGEN 310, Eppendorf, Framingham, MA, USA) for heterotrophic aseptic fermentation. The fermenter had an inner diameter of about 17.7 cm and was equipped with two Rush-ton impellers with a diameter of 7.6 cm. There were four 1.7 cm-wide baffles around the walls of the vessel. During fermentation, the pH was automatically maintained at 6.0 ± 0.05 by adding 3 M NaOH or 1 M HCl solution. The aeration rate was maintained at 1 VVM and the initial airflow rate was 2.8 L·min^−1^. The airflow rate was controlled automatically by a thermal mass flow controller, the temperature was maintained at 30 °C and dissolved oxygen (DO) was controlled automatically at 40% by the combination of the agitation speed and oxygen input. For the remaining maintenance operations during fermentation, refer to the study by Hu et al. [45]. Once the CHL in the fermenter reached a predetermined concentration, the algal liquid was subjected to spray drying (inlet temperature of 160 °C and outlet temperature of 70 °C) using a centrifugal spray dryer (LPG-500, Pioneer Drying Co., Ltd., Changzhou, China), which eventually yielded the desired CHL powder. Brown fish meal and casein protein were purchased from Guangdong Yuejia Feed Co., Ltd. (Zhanjiang, China). Corn starch, fish oil, corn oil, soybean lecithin, vitamin and mineral premix, choline chloride, antioxidants, attractant, vitamin C, calcium, cellulose, etc., were bought from Zhanjiang Kecheng Trading Co., Ltd. (Zhanjiang, China). The crude protein (CP) and crude lipids (CL) contents of brown fish meal were 68.21% and 7.8%, respectively. The CP and CL contents of casein protein were 92.30% and 0.2%, respectively. The CP and CL contents of *C. sorokiniana* powder were 57.5% and 5.5%, respectively. The diet materials were crushed and passed through 80 mesh, weighed accurately, mixed thoroughly, and then added with pre-weighed distilled water and fish oil, etc. After that, the pellets of 1.0 and 1.5 mm in diameter were prepared by a twin-screw extruder (M-256, South China University of Technology, Guangzhou, China), post-ripened in an oven at 60 °C for 30 min, dried naturally, and stored at −20 °C until use.

### 2.2. Feeding Trial

The feeding trials were conducted in an indoor culture system at the marine biology research base of Guangdong Ocean University (Zhanjiang, China). *L. vannamei* (initial mean body weight 0.83 ± 0.02 g) were purchased from Zhanjiang Yuehai Aquatic Fry Co., Ltd. (Zhanjiang, China) and temporarily reared for one week to adapt to experimental conditions before the start of the feeding trial. Six hundred *L. vannamei* were randomly selected and equally assigned to 15 fiberglass tanks (300 L) as five treatment groups with three replicates, fed by the experimental feeds four times a day (07:00, 11:00, 17:00 and 21:00) for 8 weeks under the water condition with the temperature range 20.0–30.0, salinity range 27–30 g/L, pH range 7.7–8.0, dissolved oxygen level at least 6.0 mg/L and ammonia level less than 0.05 mg/L. Each fiberglass tank was aerated individually, and the water was changed 1/3 daily.

### 2.3. Sample Collection

At the end of the 8-week farming experiment, the shrimp were starved for 24 h. Subsequently, they were weighed and counted to analyze growth performance. Twelve shrimps were randomly taken from each fiberglass tank. Three shrimps were preserved for body composition analysis the muscles from another 3 shrimps randomly selected and subsequently transferred to −80 °C refrigerator for amino acid assays. Intestines from 3 shrimps were removed with a sterile dissecting tool, the outer surface of the intestine sterilized with 75% alcohol, and then placed in liquid nitrogen for rapid freezing, followed by storage at −80 °C refrigerator for the detection of intestinal digestive enzyme activities. Intestines from another 3 shrimps were treated with the same way for the analysis of intestinal microbiota.

### 2.4. Growth Performance Analysis

Based on the recorded data, four growth performance indicators including Survival rate (SR), weight gain rate (WGR), specific growth rate (SGR) and feed conversion ratio (FCR) were calculated as follows with reference to the methods of Yadav et al. and Jayant et al. [46,47]:SR (%) = 100 × (final shrimp number/initial shrimp number);WGR (%) = 100 × (final body weight − initial body weight)/initial body weight;SGR (%) = 100 × [ln (final body weight) − ln (initial body weight)]/day;FCR = feed intake/(final body weight − initial body weight);

### 2.5. Analysis of Body Composition

The whole shrimp body composition was determined by the method of AOAC [48]. The whole shrimp were dried to constant weight at 105 °C, the crude protein content was determined by Kjeltec^TM^ 8400 (Suzhou, China), the crude lipid content was determined by Soxhlet extraction using petroleum ether as the extractant, and the ash content were determined after searing in a muffle furnace (550 °C) for 6 h. The amino acid content of muscle was determined according to the first method of GB 5009.124–2016 using a fully automatic amino acid analyzer (Hitachi L-8900, Tokyo, Japan).

### 2.6. Analysis of Intestinal Digestive Enzymes Activities

The intestinal tissues were milled and diluted with 0.9% saline at a ratio of 1:9 at 4 °C and then centrifuged at 4000 rpm for 15 min at 4 °C to extract the supernatant. The supernatants of intestinal samples were used to analyze the activities of amylase (AMS), trypsin and lipase using the kits ml036449, ml036384, and ml036371 developed by Shanghai Enzyme-linked Biotechnology Co., Ltd. (Shanghai, China). All the indicators were detected on the Thermo Scientific Microplate Reader (Thermo, Multiskan GO1510, Waltham, MA, USA) using the method according to the instructions.

### 2.7. Intestinal Microbial Analysis

The bacterial genomic DNA was extracted from intestinal samples using HiPure Soil DNA Kits (Magen, Guangzhou, China) following the manufacturer’s instructions. The obtained DNA was used for PCR using universal primers targeting the bacterial 16S rDNA gene. The primers 341F (5′-CCTACGGGNGGCWGCAG-3′) and 806R (5′-GGACTACHVGGGTATCTAAT-3′) were used to amplify V3 to V4 variable regions of the bacterial 16S rRNA gene. The negative and positive controls were set up during DNA extraction and PCR amplification. The extracted DNA products were examined by agarose gel electrophoresis. Amplification of specific regions (16s rDNA) in the DNA samples required the use of specific primers with barcode. The amplification system included 1.5 μL of primers (5 μM), 5 μL of 2.5 mMdNTPs, 5 μL of 10×KODBuffer, 1 μL of KOD polymerase and 100 ng of template DNA in a 50 μL reaction system. The amplification conditions were: pre-denaturation at 95 °C for 2 min, denaturation at 98 °C for 10 s, annealing at 62 °C for 30 s, extension at 68 °C for 30 s for 27 cycles, and finally, extension at 68 °C for 10 min. The amplification products were recovered by gel cutting and quantified by Qubit 3.0 fluorometer. Mix purified amplification products of the same mass, ligate them to sequencing adapters, and construct sequencing libraries according to Illumina’s official instructions. The PE250 pattern of the Hiseq2500 was sequenced on the computer. After raw reads were obtained by sequencing, low-quality reads were filtered and then assembled and re-filtered to ensure that the most efficient data were used to cluster into OTUs. After obtaining OTUs, species annotation, α-diversity analysis, β-diversity analysis and Tax4fun community function prediction were performed sequentially according to the analysis process. In the presence of valid groupings, differences between groups were compared and tested for differences. The sequencing and analysis were done by Guangzhou Genedenovo Biotechnology Co., Ltd. (Guangzhou, China). The raw data of the intestinal microbiota have been uploaded to NCBI SRA database (https://www.ncbi.nlm.nih.gov/sra, accessed on 17 July 2023) with the corresponding accession number: PRJNA995320.

### 2.8. Statistical Analysis

The results were expressed by mean ± standard deviation (mean ± SD), and one-way analysis of variance (one-way ANOVA) was used to test the significance for SPSS version 22, Tukey’s multiple comparison method was used to compare the data. The differences between all test results were considered significant at *p* < 0.05.

## 3. Results

### 3.1. Growth Performance

As shown in Table 2, the WGR of *L. vannamei* showed an increasing and then decreasing trend with increasing dietary protein levels (*p* < 0.05). Specifically, the WGR of in the CHL36, CHL40 and CHL44 groups were significantly higher than that in the CHL32 and CHL48 groups (*p* < 0.05), and the WGR was significantly higher in the CHL48 group than in the CHL32 group (*p* < 0.05), in addition to no significant differences between the CHL36, CHL40 and CHL44 groups (*p* > 0.05). The trend of SGR was similar to that of WGR, but there was no significant difference (*p* > 0.05). Increased dietary protein levels significantly reduced the FCR of *L. vannamei* (*p* < 0.05). Concretely, the FCR was significantly higher in the CHL32 group than in the remaining groups, and it was also significantly higher in the CHL40 and CHL44 groups than in the CHL48 group (*p* < 0.05), with no significant differences between the CHL36, CHL40 and CHL44 groups (*p* > 0.05). In addition, dietary protein level had no significant effect on SR of *L. vannamei* (*p* > 0.05). According to the regression analysis using WGR as an evaluation index, it was shown that *L. vannamei* obtained the best growth performance when the protein level in the diet was 40.81% under the condition of CHL as the main protein source (Figure 1).

### 3.2. Body Composition

As shown in Table 3, dietary protein level had significant effect on crude protein content of *L. vannamei* whole shrimp (*p* < 0.05), but not on crude lipid and ash content of whole shrimp (*p* > 0.05). When the dietary protein level was increased from 32% to 48%, the crude protein and crude lipid contents of whole shrimp first increased and then decreased and both reached a maximum at a dietary protein level of 40%. Among them, the crude protein content of CHL40 and CHL44 groups was significantly higher than that of CHL32, CHL36 and CHL48 groups (*p* < 0.05), and there was no significant difference between CHL40 and CHL44 groups as well as CHL32, CHL36 and CHL48 groups (*p* > 0.05). Similarly, the crude lipid content of whole shrimp was higher in the CHL40 group than in the remaining four groups, but there was no significant difference (*p* > 0.05).

In addition, 17 amino acids were detected in *L. vannamei* muscle, including 9 essential amino acids (EAA) and 8 non-essential amino acids (NEAA). The results showed that the amino acid content of *L. vannamei* muscle was relatively stable and to a certain extent unaffected by dietary protein levels (except for individual amino acids). Specifically, dietary protein levels had a significant effect on the content of valine (Val), histidine (His) and sum of essential amino acids (∑EAA) in *L. vannamei* muscle (*p* < 0.05). The Val and ∑EAA content in the muscles of the CHL48 group was significantly lower than that of the CHL40 group (*p* < 0.05), and its His content was significantly lower than that of the remaining groups (*p* < 0.05). It is noteworthy that the ∑EAA, sum of non-essential amino acids (∑NEAA) and sum of flavor amino acids (∑FlavorAA) contents as well as the ratio of EAA to total amino acids (TAAs) in the muscles of *L. vannamei* were all highest at a dietary protein level of 40%, but there were no significant differences (*p* > 0.05).

### 3.3. Digestive Enzyme Activities in the Intestine

As shown in Figure 2, the activities of three digestive enzymes including trypsin, lipase and AMS in the intestinal tract of *L. vannamei* showed a trend of increasing and then decreasing with the increase of dietary protein level, and all of them were significantly different (*p* < 0.05). Among them, the activity of trypsin in the CHL40, CHL44 and CHL48 groups was significantly higher than that in the CHL32 group (*p* < 0.05), but there was no significant difference among the CHL36, CHL40, CHL44 and CHL48 groups (*p* > 0.05). Lipase activity was significantly higher in the CHL40 and CHL44 groups than in the CHL48 group (*p* < 0.05), and there was no significant difference between the remaining groups (*p* > 0.05). Besides that, the AMS activity in the CHL44 group was significantly higher than the other four groups (*p* < 0.05), and there was no significant difference between the other four groups (*p* > 0.05). 

### 3.4. Intestinal Microbiota Analysis

Based on the effect of dietary protein level on the growth performance of *L. vannamei*, the intestinal microbiota of *L. vannamei* in the CHL32 group with low protein level, the CHL40 group with appropriate protein level and the CHL48 group with high protein level were analyzed. After quality control and read assembly (Table 4), all OTU with an average abundance of more than 1 were selected for Venn diagram analysis (https://omicshare.com/tools/Home/Soft/venn, accessed on 18 April 2023) when the effective label was more than 85%, so as to distinguish the same and different OTUs in the samples from different groups. As shown in Figure 3, the intestines of *L. vannamei* in the three groups shared 164 identical OTUs, and there were 285, 231 and 284 unique OTUs in the CHL32, CHL40 and CHL48 groups, respectively.

As shown in Table 5, the Simpson and Shannon indices, which reflect species richness and evenness, as well as the sobs, Chao1, and ACE indices, which reflect species richness of samples, were analyzed to assess the alpha diversity of the intestinal microbiota of *L. vannamei* in the three groups. The results showed that there was no significant differences found in Sobs, ACE, Shannon and Simpson indices among these three groups (*p* > 0.05). However, the Chao1 index of *L. vannamei* intestinal microbiota was significantly lower in the CHL48 group than that in the CHL32 and CHL40 groups (*p* < 0.05). 

At the phylum level, the top 10 species were Proteobacteria, Firmicutes, Tenericutes and Bacteroidetes. Actinobacteria, Verrucomicrobia, Cyanobacteria, Acidobacteria, Planctomycetes and Chloroflexi were the subdominant phylum (Figure 4A). As shown in Figure 4B, the relative abundance of Proteobacteria and Firmicutes as well as the ratio of Firmicutes to Bacteroidetes in the intestine of *L. vannamei* increased significantly with the increasing dietary protein levels (*p* < 0.05), reaching the highest level in the CHL48 group. The relative abundance of Firmicutes and the ratio of Firmicutes to Bacteroidetes were significantly different among the three groups (*p* < 0.05) but the relative abundance of Proteobacteria was significantly higher only in the CHL40 group than that in the other two groups (*p* < 0.05). In addition, the relative abundance of Actinobacteria were significantly different among the three groups, with the maximum in the CHL40 group (*p* < 0.05). The relative abundance of Actinobacteria in the CHL48 group was significantly lower than the CHL40 group and significantly higher than the CHL32 group (*p* < 0.05).

As shown in Figure 5A, the top 10 species of the intestinal microbiota *L. vananmei* at the family level were Vibrionaceae, Mycoplasmataceae, Erysipelotrichaceae, Flavobacteriaceae, Enterobacteriaceae, Moraxellaceae, Rhodobacteraceae, Leuconostocaceae, Rubritaleaceae and Cymbidium_faberi. The relative abundance of Vibrionaceae and Flavobacteriaceae in the intestine of *L. vannamei* decreased significantly and the relative abundance of Rhodobacteraceae increased significantly with the increase of dietary protein level (*p* < 0.05). Among them, the relative abundances of Flavobacteriaceae were significantly different among all the three groups (*p* < 0.05). The relative abundance of Vibrionaceae and Rhodobacteraceae were highest in the CHL32 and CHL48 groups, respectively (*p* < 0.05), which were significantly higher than that in the other two groups (Figure 5B).

At the genus level, *Vibrio*, *Candidatus_Bacilloplasma*, ZOR0006, *Acinetobacter* and *Escherichia-Shigella* were the major dominant genera. *Spongiimonas*, *Weissella*, *Shimia*, *Ruegeria* and *Leuconostoc* were the second dominant genera (Figure 6A). As shown in Figure 6B, the relative abundance of *Vibrio* and *Spongiimonas* significantly decreased while the relative abundance of *Ruegeria* significantly increased with the increasing dietary protein levels (*p* < 0.05). The relative abundance of *Vibrio* was not significantly different between the CHL32 and CHL40 groups but both of them were significantly higher than that in the CHL48 group. The relative abundance of *Ruegeria* was not significantly different between the CHL40 and CHL48 groups but both of them were significantly higher than that in the CHL32 group. Besides, the relative abundances of *Spongiimonas* were significantly different among all the three groups (*p* < 0.05).

The functions of the *L. vannamei* intestinal microbiota were predicted using Tax4fun version 2020, and a total of 37 categories of functions including carbohydrate metabolism, amino acid metabolism, metabolism of cofactors and vitamins were predicted (Figure 7). The top 10 predicted functions in relative abundance were carbohydrate metabolism (relative abundance 13.57%), amino acid metabolism (relative abundance 11.82%), metabolism of cofactors and vitamins (relative abundance 6.87%), functions related to energy metabolic signalling (relative abundance 6.63%), nucleotide metabolism (relative abundance 5.33%), xenobiotics biodegradation and metabolism (relative abundance, 3.75%), lipid metabolism (relative abundance 3.58%), metabolism of terpenoids and polyketides (relative abundance 2.82%), metabolism of other amino acids (relative abundance 2.76%) and glycan biosynthesis and metabolism (relative abundance 2.54%), respectively (Figure 8A). Welch’s *t*-test showed that Nicotinate and nicotinamide metabolism, Biotin metabolism and RNA transport-related functions were significantly higher in *L. vananmei* in the CHL40 group when compared to the CHL48 group at KEGG pathway level 3 (Figure 8B).

## 4. Discussion

In the face of the growing demand for animal proteins for human consumption, the development of aquaculture would undoubtedly be a good solution [49]. However, the dramatic increase in the use of FM is the current bottleneck in the sustainable development of aquaculture. It is generally accepted that replacing FM in aquaculture diets with sufficient new protein sources, such as CHL, could maintain the rapid growth of aquatic animals [50,51,52,53]. However, because aquatic animals differ in their ability to utilise proteins from different sources, the optimal dietary protein levels differ when utilising different protein sources [54,55]. Studies on the effects of different protein diets on shrimp are relatively rare. Therefore, it is important to improve understanding of the dietary protein requirements of shrimp when different protein sources are used. In this study, the growth performance, body composition, intestinal digestion and microbiota of *L. vannamei* fed by diets with different protein levels (using CHL as the main protein source) were investigated, to investigate the practicability of using CHL in the feed of shrimp as an FM substitute. 

Consistent with studies on *Ancherythroculter nigrocauda* [56], *Oncorhynchus mykiss* [57], *Larimichthys polyactis* [58] and *Totoaba macdonaldi* [59], the WGR of *L. vannamei* in the present study showed an increasing and then decreasing trend with increasing dietary protein levels, while the SR was not affected. This may have been because all animals have optimum protein requirements, so dietary protein levels that are too high or too low are detrimental to their growth. For *L. vannamei*, specifically, too low dietary protein levels (32%) resulted in inadequate energy intake, which was used primarily to maintain basic life activities rather than for growth [60,61,62]; in contrast, excessive dietary protein levels (48%) increased the metabolic cost of nitrogen excretion, with the portion of protein that could not be digested becoming a metabolic burden and causing metabolic disturbances, ultimately limiting the growth of *L. vannamei* [63,64]. Often, these negative effects only restricted the growth of *L. vannamei* but were not directly life-threatening. This also explained the significantly lower WGR of *L. vannamei* in the CHL32 and CHL48 groups compared to the middle three groups, while the SR did not change significantly. Additionally, the regression analysis using WGR as an evaluation index found that 40.81% was the optimal dietary protein level for *L. vannamei* when CHL was the main protein source. Furthermore, consistent with the results of studies on *Bidyanus bidyanus* [65], *Scophthalmus maximus* [66] and *Dicentrarchus labrax* [67], elevated dietary protein levels were found to significantly reduce the FCR in *L. vannamei*. This may have been because aquatic animals can partially compensate for the effects of low protein content in their diet by increasing their voluntary intake of the diet [68,69].

Apart from the differences in growth performance, the present study also revealed differences in the body composition of *L. vannamei*. The crude protein content of *L. vannamei* whole shrimp showed an increasing and then decreasing trend with increasing dietary protein levels, reaching a maximum near the CHL40 and CHL44 groups. These results were consistent with those obtained for *Larimichthys polyactis* [58], *Trachinotus ovatus* [70], *Dentex dentex* [71] and *Pseudobagrus fulvidraco* [72]. Typically, when dietary protein levels are increased within an appropriate range, the intake and digestibility of protein in farmed animals increase; that is, more dietary protein is consumed, digested and absorbed, which is converted into building block proteins and ultimately used in the generation and repair of body tissues [73,74]. However, when dietary protein levels exceed the appropriate range, the additional protein cannot be digested and absorbed by the farmed animals. This portion of the protein burdens the digestive and metabolic systems of farmed animals, preventing them from functioning efficiently and ultimately resulting in a relative reduction in their crude protein content [75]. The present study revealed that the crude lipid content of *L. vannamei* whole shrimp changed insignificantly but roughly similarly to crude protein content as the dietary protein levels increased. This is consistent with findings in *L. polyactis* [58] and *Ctenopharyngodon idellus* [74]. It is commonly believed that, as dietary protein levels increase, part of the protein is converted to lipids and deposited in an animal’s body, thus contributing to the increased body lipid content of farmed animals [76]. However, other studies have shown that diets with lower protein-to-energy ratios may exert a high-energy effect, which also leads to the accumulation of body lipids in farmed animals; this would mean that farmed animals consuming high-protein diets have lower body lipid content [77]. It has also been shown that dietary protein levels do not affect the body lipid content of farmed animals [78]. The reasons underpinning these different experimental results may need further exploration, but the present study clearly revealed that a dietary protein level of about 40% resulted in the optimal protein and lipids deposition rates for *L. vannamei* when CHL was the main protein source.

Protein content is the key index for measuring the quality of aquatic animal muscle [79] and the amino acid composition is the key factor to measure the protein quality [80]. Both the protein content and amino acid composition affect the nutritional value of aquatic animals. In the present study, although the content of various amino acids in the muscle tissue of *L. vannamei* was relatively stable and largely unaffected by dietary protein levels, several indices including ∑EAA, ∑NEAA and ∑FlavourAA content and the ratio of EAA to TAAs reached their highest values when the dietary protein level was 40%. FlavourAA relates to the source of the delicious flavour in shrimp muscle [81], EAA plays an important role in maintaining human health, and the EAA content and the ratio of EAA to TAAs determine the nutritional value of muscle protein in shrimp to humans [82]. Thus, the improvement in these metrics implied that 40% dietary protein may result in better muscle quality in *L. vannamei*.

Digestive enzymes facilitate the digestive action of the animal body [83]. Improving digestive enzyme activity can effectively promote shrimp digestion and their absorption of nutrients, thus promoting shrimp growth [84,85,86]. It has been shown that the digestive enzyme activity of shrimp varies with dietary composition [87]. In *Procambarus clarkii*, the activity of various digestive enzymes including trypsin, lipase and AMS in the hepatopancreas and intestine showed a tendency of increasing and then decreasing as the dietary protein level increased, with significant differences in the activity of trypsin and lipase [88]. Similar changes in digestive enzyme activity were found in the present study, which was also consistent with the changes in growth performance and body composition. Based on these results, it was concluded that increasing dietary protein levels within a certain range effectively improved the digestion of *L. vannamei*, which in turn promoted the deposition of proteins and lipids, and ultimately, growth. However, excessive dietary protein levels will burden the digestive system [89] and result in inhibitory rather than facilitative effects.

Intestinal microbiota plays an important role in maintaining the health of an organism [90,91,92,93]. It is generally accepted that the type of protein added to the diet is closely related to the composition and function of the intestinal microbiota, with low-quality protein sources increasing the relative abundance of intestinal pathogenic bacteria and intestinal dysfunction [94,95,96,97,98]. However, research on the effect of protein levels on the intestinal microbiota of aquatic animals is still lacking. In this study, the effect of different protein levels on the intestinal microbiota of *L. vannamei* was investigated. The results showed that the Chao1 index—which reflects the species richness of intestinal microbiota—was significantly lower for the CHL48 group than for the CHL40 group, whereas this was not the case for the CHL32 group. This indicated that low protein levels had less effect on the diversity and abundance of the intestinal microbiota community of *L. vannamei* than high protein levels when CHL was used as the main protein source, which was consistent with the findings of Zhao et al. [99].

Additionally, changes in dietary protein levels also had significant effects on the intestinal microbiota composition at different levels in *L. vannamei*. Generally, elevated dietary protein levels significantly increased the relative abundance of beneficial microbiota while also leading to a significant decrease in pathogenic bacteria at the phylum, family and genus levels. At the phylum level, the relative abundance of Proteobacteria and Firmicutes in the intestine of *L. vannamei* increased significantly with increasing dietary protein levels. Additionally, the relative abundance of Rhodobacteraceae and *Ruegeria* was significantly elevated while the relative abundance of Vibrionaceae, *Flavobacterium* and *Spongiimonas* was significantly decreased. An increase in Firmicutes increases the number of lipid droplets, which proportionally intensifies the absorption of fatty acids by the host organism [100]. Most members of the Rhodobacteraceae can produce vitamin B12, which safeguards the growth of shrimp [101,102]. *Ruegeria*, a flagellate-associated bacterium, is considered to be beneficial due to its potential role in protecting aquatic organisms against pathogenic *Vibrio* species [103,104]. Vibrionaceae is the most common core group of bacteria in the intestinal tract of crustaceans and many members of this family are conditional pathogens that can have serious effects on the survival and health of crustaceans [105,106]. Both *Flavobacterium* and *Spongiimonas* belong to Bacteroidetes, which contain multiple pathogenic factors and are conditional pathogens [107,108]. Firmicutes and Bacteroidetes are interesting intestinal microbes that are usually the dominant microbiota in the human and other mammalian intestines and are involved in the digestion and metabolism of food by the host [109]. In aquatic animals, these phyla are usually considered to help improve digestive capacity. In the present study, the ratio of Firmicutes to Bacteroidetes in the intestinal tract of *L. vannamei* increased as protein levels increased. In combination with the changes in growth performance and digestive enzyme activity, these results indicated that an appropriate protein level was beneficial for intestinal health and could improve the feed utilisation of *L. vannamei*.

The Tax4fun functional prediction of intestinal microbiota showed that the top 10 significantly enriched functions were all related to metabolism, including carbohydrate metabolism, amino acid metabolism and the metabolism of cofactors and vitamins. Additionally, nicotinate and nicotinamide metabolism and biotin metabolism functions were significantly higher in the CHL40 group compared to the CHL48 group. Nicotinate and nicotinamide metabolism is associated with a variety of energy metabolic activities and may be closely related to obesity in humans [110,111]. Biotin functions as a cofactor for several carboxylase enzymes with key roles in metabolism. A study in pregnant women showed that increased metabolism accelerated biotin metabolism [112]. Thus, the results of the present study implied that an optimum protein level may be more favourable for the metabolic activity of *L. vannamei* compared to high protein levels, which was also consistent with the changes in digestive enzyme activity.

## 5. Conclusions

Taken together, the results of this study suggested that dietary protein levels of around 40% resulted in the optimal growth performance, nutrient composition, digestibility and intestinal microbiota stability of *L. vannamei* when *C. sorokiniana* was used as the main protein source. Additionally, the intestinal microbiota analysis showed that higher dietary protein levels could effectively decrease pathogenic bacteria and increase beneficial bacteria and inhibited metabolic activity, simultaneously. This study helps address the former lack of research on optimal dietary protein levels for *L. vannamei*, with the creative use of intestinal microbiota sequencing technology as an evaluation index and the introduction of *C. sorokiniana* as the main protein source. The results of the study have important implications for sustainable and standardised *L. vannamei* farming.

## Figures and Tables

**Figure 1 animals-13-02881-f001:**
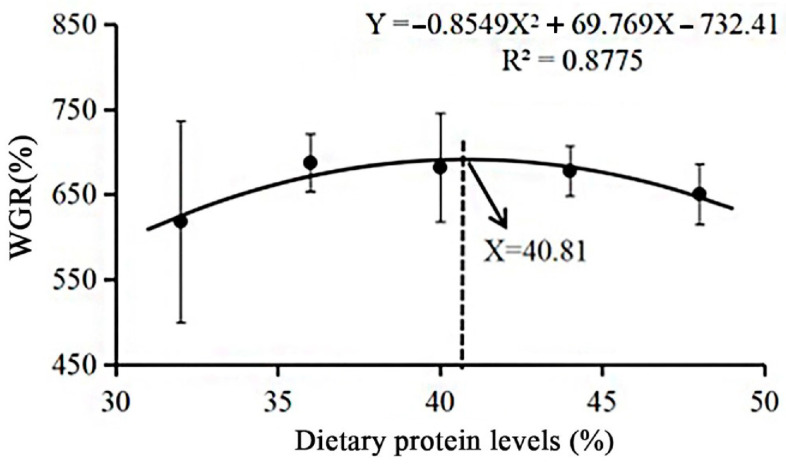
The relationship between dietary protein level and WGR. A regression model was used to fit a functional relationship between protein levels and WGR.

**Figure 2 animals-13-02881-f002:**
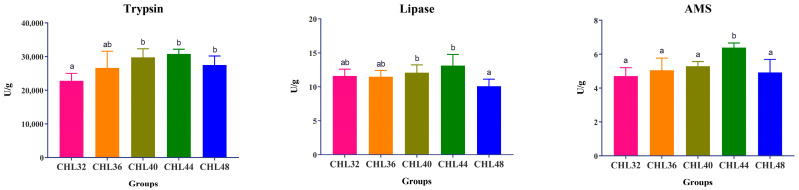
Effect of dietary protein level on the activities of intestinal digestive enzymes in *L. vannamei*. Different superscript letters indicate significant differences exist among treatments (*p* < 0.05).

**Figure 3 animals-13-02881-f003:**
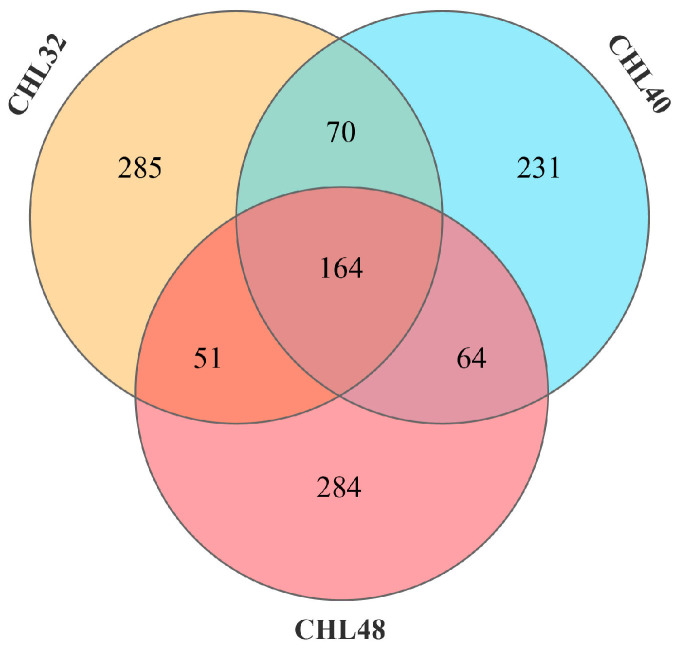
OTUs Venn diagram of the intestinal microbiota in *L. vannamei*. Overlapping regions are OTUs shared between two or three groups, and non-overlapping regions are OTUs specific to each group.

**Figure 4 animals-13-02881-f004:**
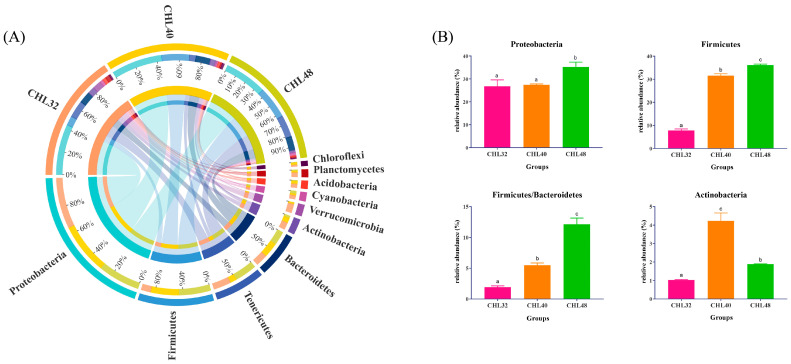
Intestinal microflora composition at the phylum level in *L. vannamei*. (**A**) Mean abundance in different groups. One side of the graph is the grouping information, and the other side is the species information. The lines on both sides represent corresponding relationship pairs. The thicker the lines, the greater the abundance value. (**B**) Relative abundance of dominant phylum with significant differences. Different letters represent significant differences between groups (*p* < 0.05).

**Figure 5 animals-13-02881-f005:**
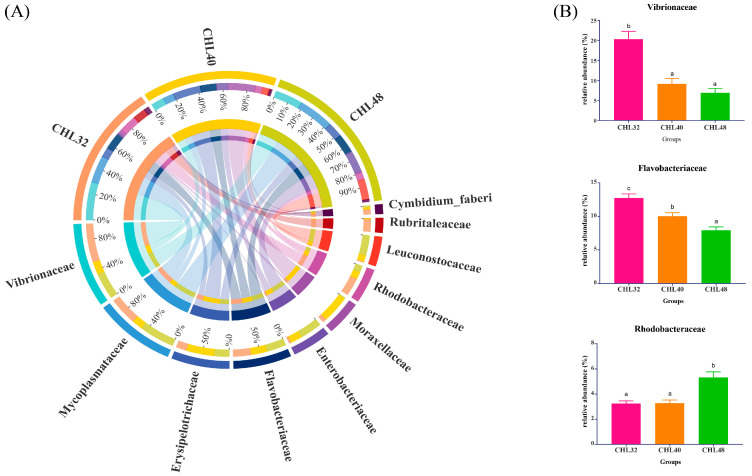
Intestinal microflora composition at the family level in *L. vannamei*. (**A**) Mean abundance in different groups. One side of the graph is the grouping information, and the other side is the species information. The lines on both sides represent corresponding relationship pairs. The thicker the lines, the greater the abundance value. (**B**) Relative abundance of dominant families with significant differences. Different letters represent significant differences between groups (*p* < 0.05).

**Figure 6 animals-13-02881-f006:**
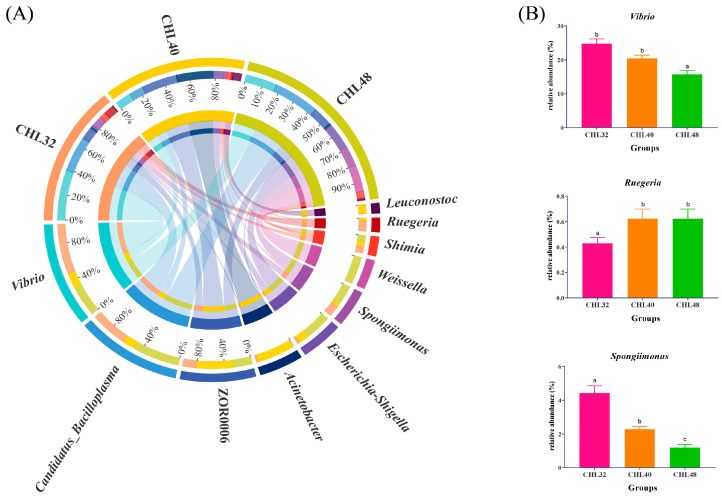
Intestinal microflora composition at the genus level in *L. vannamei*. (**A**) Mean abundance in different groups. One side of the graph is the grouping information, and the other side is the species information. The lines on both sides represent corresponding relationship pairs. The thicker the lines, the greater the abundance value. (**B**) Relative abundance of dominant genus with significant differences. Different letters represent significant differences between groups (*p* < 0.05).

**Figure 7 animals-13-02881-f007:**
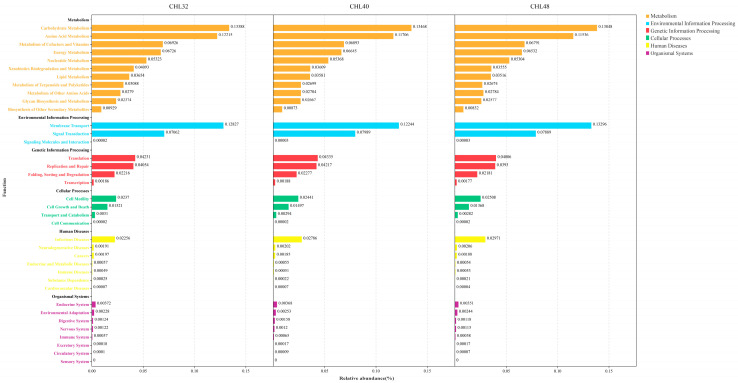
Overview of the functional distribution of KEGG in the intestinal microbiota of *L. vannamei*. Using the KEGG pathway database, which is a metabolic pathway database, biological metabolic pathways are classified into seven categories: Metabolism, Genetic Information Processing, Environmental Information Processing, Cellular Processes, Organismal Systems, Human Diseases, and Drug Development. The predicted functions in the major classes were ranked in order of relative abundance.

**Figure 8 animals-13-02881-f008:**
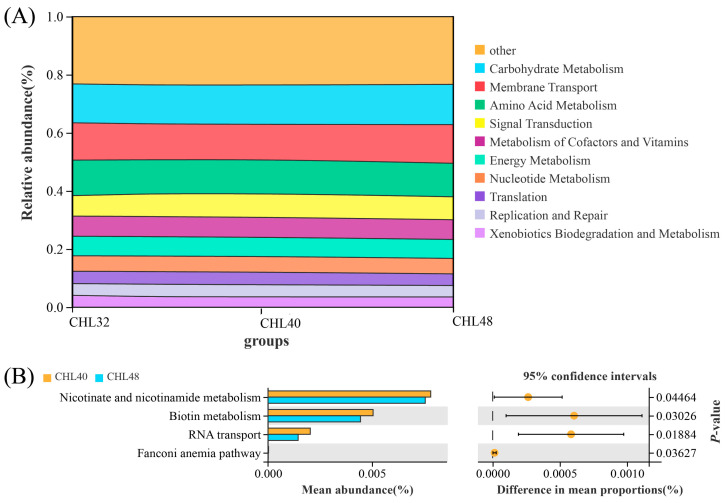
Functional prediction of dietary protein levels in the intestinal microbial community. (**A**) Relative abundances of the top 10 predicted functions. (**B**) Welch’s *t*-tests of the significantly different functions at level 3.

**Table 1 animals-13-02881-t001:** The formula and proximate composition of the basal diet (dry matter basis, g/kg).

Ingredients	Groups				
	CHL32	CHL36	CHL40	CHL44	CHL48
Brown fish meal	150	150	150	150	150
Casein protein	80	80	80	80	80
Corn starch	150	150	150	150	150
*C. sorokiniana* powder	251	321	390	460	530
Fish oil	20.2	18.25	16.35	14.45	12.5
Corn oil	20.2	18.25	16.35	14.45	12.5
Soybean lecithin	10	10	10	10	10
Vitamin and Mineral premix ^a^	12	12	12	12	12
Choline chloride	5	5	5	5	5
Antioxidants	0.5	0.5	0.5	0.5	0.5
Attractant	1	1	1	1	1
Vitamin C	0.5	0.5	0.5	0.5	0.5
Calcium	12	12	12	12	12
Cellulose	287.6	221.5	156.3	90.1	24
Total	1000	1000	1000	1000	1000
Nutrient composition (DM, %)					
Crude protein	33.02	37.98	41.37	45.51	48.92
Crude lipids	6.77	7.59	7.66	7.95	7.45
Ash	7.15	7.56	7.11	7.03	8.21
Dry matter	91.56	92.92	91.52	91.24	89.74

Note: ^a^ Vitamin and mineral premixes (per kg diet): Vitamin B1, 25 mg; Vitamin B2, 45 mg; Vitamin B3, 60 mg; Vitamin B5, 200 mg; Vitamin B6, 20 mg; Vitamin B7, 1.20 mg; Vitamin B12, 0.1 mg; Inositol, 800 mg; Folic acid, 20 mg; Vitamin A, 32 mg; Vitamin E, 120 mg; Vitamin D3, 5 mg; Vitamin K3, 10 mg. Sodium fluoride, 2 mg; Potassium iodide, 0.8 mg; Cobalt chloride (%), 50 mg; Cupric sulphate, 10 mg; Ferrous sulphate, 80 mg; Zinc sulphate, 50 mg; Manganese sulphate, 60 mg; Magnesium sulfate, 1200 mg; Sodium chloride, 100 mg; Zeolite powder, 1447.2 mg.

**Table 2 animals-13-02881-t002:** Effect of different protein levels in diets on the growth performance of *L. vannamei*.

Indexes	Groups				
	CHL32	CHL36	CHL40	CHL44	CHL48
IBW (g)	0.83 ± 0.02	0.83 ± 0.02	0.83 ± 0.02	0.83 ± 0.02	0.83 ± 0.02
FBW (g)	5.96 ± 0.04 ^a^	6.54 ± 0.03 ^b^	6.48 ± 0.20 ^b^	6.46 ± 0.16 ^b^	6.23 ± 0.03 ^ab^
WGR (%)	618.21 ± 3.14 ^a^	687.29 ± 5.01 ^c^	681.90 ± 4.60 ^c^	677.67 ± 6.79 ^c^	650.46 ± 17.55 ^b^
SGR (%/day)	3.52 ± 0.09	3.68 ± 0.08	3.67 ± 0.15	3.65 ± 0.23	3.60 ± 0.08
FCR	1.73 ± 0.04 ^c^	1.54 ± 0.01 ^ab^	1.69 ± 0.02 ^b^	1.55 ± 0.08 ^b^	1.48 ± 0.06 ^a^
SR (%)	97.5 ± 2.5	99.17 ± 1.44	97.5 ± 2.5	98.33 ± 1.44	98.33 ± 1.81

Note: Data were presented as mean ±  SD, *n*  =  3. No superscript or the same superscript in the same line means no significant difference (*p* > 0.05), and values with different superscripts in the same row differ significantly (*p* < 0.05).

**Table 3 animals-13-02881-t003:** Effect of different protein levels in diets on whole shrimp body composition and muscle amino acid composition of *L. vannamei*.

Indexes	Groups				
	CHL32	CHL36	CHL40	CHL44	CHL48
Whole body composition					
Crude protein (%)	73.02 ± 0.62 ^a^	73.35 ± 0.62 ^a^	74.73 ± 0.44 ^b^	74.46± 0.62 ^b^	73.45± 0.29 ^a^
Crude lipid (%)	7.17 ± 0.25	7.60 ± 0.40	8.18 ± 0.38	7.96 ± 0.55	7.37 ± 0.31
Ash (%)	12.18 ± 0.97	12.63 ± 1.00	11.60 ± 1.10	13.87 ± 1.85	12.37 ± 0.71
Essential amino acids					
Thr	0.76 ± 0.05	0.72 ± 0.02	0.79 ± 0.03	0.73 ± 0.03	0.74 ± 0.04
Val	0.82 ± 0.03 ^ab^	0.81 ± 0.03 ^ab^	0.87 ± 0.06 ^b^	0.78 ± 0.04 ^ab^	0.75 ± 0.04 ^a^
Met	0.59 ± 0.05	0.57 ± 0.02	0.58 ± 0.04	0.57 ± 0.06	0.52 ± 0.03
Ile	0.8 ± 0.04	0.77 ± 0.05	0.82 ± 0.06	0.74 ± 0.02	0.71 ± 0.03
Leu	1.55 ± 0.06	1.49 ± 0.05	1.55 ± 0.07	1.46 ± 0.1	1.36 ± 0.05
Phe	0.84 ± 0.05	0.81 ± 0.01	0.84 ± 0.05	0.79 ± 0.06	0.73 ± 0.01
Lys	1.67 ± 0.09	1.62 ± 0.04	1.7 ± 0.1	1.61 ± 0.1	1.49 ± 0.07
His	0.43 ± 0.01 ^b^	0.4 ± 0.03 ^b^	0.42 ± 0.01 ^b^	0.40 ± 0.01 ^b^	0.34 ± 0.0 ^a^
Arg	2.01 ± 0.26	2.09 ± 0.13	2.29 ± 0.08	2.19 ± 0.06	2.00 ± 0.23
Non-essential amino acids					
Asp	2.05 ± 0.08	1.98 ± 0.05	2.09 ± 0.1	1.96 ± 0.08	1.88 ± 0.09
Glu	3.39 ± 0.11	3.33 ± 0.09	3.4 ± 0.15	3.22 ± 0.22	3.11 ± 0.14
Ala	1.94 ± 0.05	2.06 ± 0.03	2.08 ± 0.16	2.07 ± 0.1	1.95 ± 0.06
Gly	1.37 ± 0.04	1.31 ± 0.07	1.39 ± 0.09	1.33 ± 0.02	1.32 ± 0.1
Serine	0.76 ± 0.05	0.72 ± 0.02	0.79 ± 0.03	0.73 ± 0.03	0.70 ± 0.04
Tyr	0.83 ± 0.13	0.80 ± 0.06	0.80 ± 0.05	0.81 ± 0.12	0.70 ± 0.03
Pro	1.16 ± 0.05	1.07 ± 0.03	1.11 ± 0.03	0.96 ± 0.17	0.96 ± 0.15
Total amino acids (TAAs)	20.93 ± 0.87	20.57 ± 0.21	21.5 ± 1.04	20.37 ± 1.06	19.23 ± 0.76
∑EAA	9.46 ± 0.53 ^ab^	9.28 ± 0.07 ^ab^	9.85 ± 0.5 ^b^	9.28 ± 0.46 ^ab^	8.61 ± 0.46 ^a^
∑NEAA	11.49 ± 0.36	11.28 ± 0.18	11.64 ± 0.54	11.08 ± 0.61	10.64 ± 0.32
EAA/TAAs	45.20	45.11	45.81	45.56	44.77
∑FlavorAA	8.75 ± 0.15	8.69 ± 0.19	8.97 ± 0.5	8.58 ± 0.28	8.26 ± 0.28

Note: The whole shrimp body composition was dry matter basis (%), and muscle amino acids were wet matter basis (%). Data were presented as mean ±  SD, n  =  3. No superscript or the same superscript in the same line means no significant difference (*p* > 0.05), and values with different superscripts in the same row differ significantly (*p* < 0.05). TAAs: sum of total tested amino acids; ∑EAA: sum of essential amino acids, including threonine (Thr), valine (Val), methionine (Met), isoleucine (Ile), leucine (Leu), phenylalanine (Phe), lysine (Lys), histidine (His) and arginine (Arg); ∑NEAA: sum of non-essential amino acids, including aspartic acid (Asp), glutamic acid (Glu), alanine (Ala), glycine (Gly), serine (Ser), tyrosine (Tyr) and proline (Pro); EAA/TAAs: the percentage of EAA to TAAs; ∑FlavorAA: sum of flavor amino acids, including aspartic acid, glutamic acid, glycine and alanine.

**Table 4 animals-13-02881-t004:** Read quality assessment of intestinal microbiota sequencing in *L. vannamei*.

Sample Name	Indexes						
	Raw PE	Clean PE	Raw Tags	Clean Tags	Chimera	Effective Tags	Effective Ratio (%)
CHL32-1	122,664	122,022	106,842	105,726	446	105,280	85.83
CHL32-2	120,260	119,749	110,074	107,945	4668	103,277	85.88
CHL32-3	120,728	120,167	109,686	108,565	571	107,994	89.45
CHL40-1	130,754	130,212	119,650	117,333	5527	111,806	85.51
CHL40-2	135,666	135,206	126,534	125,580	1125	124,455	91.74
CHL40-3	129,857	129,236	116,454	115,410	497	114,913	88.49
CHL48-1	129,159	128,678	119,762	118,160	1996	116,164	89.94
CHL48-2	130,380	129,817	119,996	119,073	5492	113,581	87.12
CHL48-3	126,165	125,662	115,396	114,278	1587	112,691	89.32

Note: Raw PE, Raw Pair-end Reads; Clean PE, Clean Pair-end Reads.

**Table 5 animals-13-02881-t005:** Statistics of the alpha diversity indexes of the intestinal microbiota of *L. vannamei*.

Indexes	Groups		
	CHL36	CHL40	CHL44
Sobs	433.33 ± 18.17	430.33 ± 12.09	457.67 ± 17.01
ACE	723.39 ± 2.45	744.53 ± 26.78	734.80 ± 4.63
Chao1	758.89 ± 2.47 ^b^	769.60 ± 13.91 ^b^	647.45 ± 16.56 ^a^
Shannon	3.79 ± 0.12	4.04 ± 0.76	4.04 ± 0.36
Simpson	0.85 ± 0.04	0.91 ± 0.02	0.83 ± 0.04

Note: Data were presented as mean ±  SD, n  =  3. No superscript or the same superscript in the same line means no significant difference (*p* > 0.05), and values with different superscripts in the same row differ significantly (*p* < 0.05).

## Data Availability

The data that support the findings of this study are available on request from the corresponding author. The data is not publicly available due to privacy or ethical restrictions.

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
