# Peer review of "The Effects of Dietary Protein Level on the Growth Performance, Body Composition, Intestinal Digestion and Microbiota of Litopenaeus vannamei Fed Chlorella sorokiniana as the Main Protein Source"

_animals, 2023, doi:10.3390/ani13182881_

Round 1

Reviewer 1 Report (Previous Reviewer 1)

The authors unfortunately ignored some previous comments and recommendations. The authors need to improve their work.

1. The introduction does not adequately disclose the selection and characteristics of chlorella microalgae as an alternative protein source for the reduction or replacement of fish meal in aquaculture feed. The authors only briefly touched on modes of microalgae cultivation. This is very important because, depending on what medium (synthetic media or alternative cheap feedstocks available today, such as industrial wastes, agricultural wastes and residues, and aquaculture wastes….) and in what system the algal biomass will be obtained, the economic and ecological sides of biotechnology will depend.

So, authors should disclose to readers the bottlenecks of microalgae-assisted aquaculture and strategies to improve the economic feasibility and sustainability of microalgae cultivation for future applications. I would advise to present the main provisions of the reviews and the data of experimental works carried out in this area in the introduction/discussion:

https://doi.org/10.1016/j.rser.2022.112284,

https://doi.org/10.3390/biotech12030053,

https://doi.org/10.3390/plants10030478,

https://doi.org/10.3390/app9112377,

https://doi.org/10.1007/s12649-018-0256-3,

https://doi.org/10.1016/j.biortech.2019.121625.

2. The authors need to note the parameters for algae cultivation for both inoculum and biomass production (temperature, pH level, volume of medium, aeration rate, agitation speed, etc.) and give in detail the preparation of the algal powder (details of filtration, dehydration, washing, concentration, wall-breaking, and drying steps). Note for readers that a heterotrophic regimen has been applied, provide a reference, or indicate the composition of the modified Endo medium. Have manipulations been carried out to reduce the contamination of microalgae cultures?

3. Table 4. Raw/ Clean PE – give full names.

4. Simple Summary and Conclusions are similar. Please rephrase.

Author Response

Reviewer 2 Report (Previous Reviewer 2)

As the article is resubmission of the same article and author has improved alot however some technical queries still persists so those needs to be clarified and rest of the comments are attatched as review report

Line no.35-36 is scientifically and logically contradictory as proximate composition of an individual is almost constant that should not be affected by the external factors including dietary application of a feed additives. It needs proper justification. 

Conclusion need to refine and it should be prospective not as such results brief.

Language required a substantial revision in terms of grammar, phrases and sentence framing.

a through revision is required for the article

Author Response

Reviewer 3 Report (Previous Reviewer 3)

there only slight modifications that could be made before final acceptance

maybe a few graphics could be removed?

good

Author Response

请参阅附件。

Round 2

Reviewer 1 Report (Previous Reviewer 1)

LL. 151-173. Please replace "fermentation" with "cultivation".

-

Reviewer 2 Report (Previous Reviewer 2)

The article now can be accepted for publication

Now it is ok.

This manuscript is a resubmission of an earlier submission. The following is a list of the peer review reports and author responses from that submission.

Round 1

Reviewer 1 Report

The work under review is relevant because today's promising aquaculture industry has certain limitations and requires rapid and healthy development. The authors highlighted the importance of finding aquafeed protein sources and included microalgae in the list of promising protein sources for future aquafeeds. However, in the Introduction, there is no substantiation that the cultivation of Chlorella, namely C. sorokiniana, is possible on different media, including cheap media, and that microalgae can grow under various cultivation modes, and many strains are highly adaptive. I would advise you to cite these sources of literature in this area: doi: 10.1371/journal.pone.0199873; doi.org/10.3390/biology9070169; DOI: 10.1038/s41598-018-24979-8; https://doi.org/10.3390/biology7020025...and others.

Please, see specific and general comments below for details:

1. In materials and methods, there is no data on microalgae and the mode of their cultivation. How was algal powder obtained? After all, the conditions for cultivating microalgae will be reflected in the total costs.

 2. Table 1: In what units of measurement is the composition? Why is the proportion of cellulose different in treatments?

 3. Section 2.7: What kit was used for total DNA extraction? And indicate what quality of reads were evaluated in the work.

Rephrase the sentence: The bacterial DNA from the intestine were extracted and amplified using primers for the V3+V4 region of the bacterial 16S rDNA gene (341F: 5'-CCTACGGGNGGCWGCAG-3', 806R: 5'-GGACTACHVGGGTATCTAAT-3'). For example: The obtained DNA was used for PCR using universal primers targeting the bacterial 16S rRNA gene. The primers 341F (5'-CCTACGGGNGGCWGCAG-3') and 806R (5'-GGACTACHVGGGTATCTAAT-3') were used to amplify V3 to V4 variable regions of the bacterial 16S rRNA gene.

Were negative controls supplied for DNA extraction and PCR? Give a link to the program that was used to build the Venn diagram.

4. Section 3.4: Give a link to a project registered in one of the databases.

5. Increase the readability of figures 2, 4B, 5B, 6B and 7 (the legend and axes).

6. In the conclusions section, give more specific conclusions about the work.

-

Reviewer 2 Report

Title:The effects of dietary protein level on the growth performance, body composition, intestinal digestion and microbiota of Litopenaeus vannamei fed Chlorella sorokiniana as the main protein source
Simple Summary
It need a through rewriting for an effective flow of reading.
Abstract
I have a query that is for Litopenaeus vannamei the protein requirement is known then what is novelty of this work.
Authors have used 5 diets including positive control then rather mentioning 40 % by drawing polynomial regression the optimum level can be calculated so this is to be given in revised MS.
Introduction
Line 96-99 its not related to the present study. it needs a clear-cut sentence.
2. Materials and Method
Table no 1 why author has taken so much of cellulose is it digestible????????????? Need to explain
2.4. Growth performance analysis
Author have used very few indices and even not given any references for that I would suggest author to give following references
Geetanjali Yadav, Dharmendra Kumar Meena, Amiya Kumar Sahoo, Basanta Kumar Das, Ramkrishna Sen,Effective valorization of microalgal biomass for the production of nutritional fish-feed supplements, Journal of Cleaner Production,Volume 243,2020,118697
M. Jayant, M.A. Hassan, P.P. Srivastava, D.K. Meena, P. Kumar, A. Kumar, M.S. Wagde, Brewer’s spent grains (BSGs) as feedstuff for striped catfish, Pangasianodon hypophthalmus fingerlings: An approach to transform waste into wealth, Journal of Cleaner Production, Volume 199, 2018,Pages 716-722.
Results and discussion
Lack of coherence between results and discussing so need to revise thoroughly these two sections

there are some gramatical and phrase related error were also observed

Reviewer 3 Report

there is some originality in the present work.and the concern on microbiota is a good point, therefore the short length of intestine indicate a rapid transit time and the presence/absence of cellulose can possibly effect on transit time. a question could be raised on.

can be slightly improved and solve the question of cellulose content >20% that reached in one experimental diet a substantial level.

a proof reading would be necessary. what about taurine?

can be slightly improved and solve the question of cellulose content >20% that reached in one experimental diet a substantial level.
